# Harnessing the Power of Zinc-Solubilizing Bacteria: A Catalyst for a Sustainable Agrosystem

Swapnil Singh, Rohit Chhabra , Ashish Sharma * and Aditi Bisht *

Department of Botany and Environment Science, DAV University, Jalandhar 144001, India; swapnbrij@gmail.com (S.S.); rohitchhabra325.rr@gmail.com (R.C.)
* Correspondence: ashish10210@davuniversity.org (A.S.); bishtaditi1994@gmail.com (A.B.)

**Abstract:** A variety of agrochemicals, especially fertilizers, are applied indiscriminately by farmers across trapezoidal landscapes to increase productivity and satisfy the rising food demand. Around one-third of the populace in developing nations is susceptible to zinc (Zn) deficiency as a result of their direct reliance on cereals as a source of calories. Zinc, an essential micronutrient for plants, performs several critical functions throughout the life cycle of a plant. Zinc is frequently disregarded, due to its indirect contribution to the enhancement of yield. Soil Zn deficiency is one of the most prevalent micronutrient deficiencies that reduces crop yield. A deficiency of Zn in both plants and soils results from the presence of Zn in fixed forms that are inaccessible to plants, which characterizes the majority of agricultural soils. As a result, alternative and environmentally sustainable methods are required to satisfy the demand for food. It appears that the application of zinc-solubilizing bacteria (ZSB) for sustainable agriculture is feasible. Inoculating plants with ZSB is likely a more efficacious strategy for augmenting Zn translocation in diverse edible plant components. ZSB possessing plant growth-promoting characteristics can serve as bio-elicitors to promote sustainable plant growth, through various methods that are vital to the health and productivity of plants. This review provides an analysis of the efficacy of ZSB, the functional characteristics of ZSB-mediated Zn localization, the mechanism underlying Zn solubilization, and the implementation of ZSB to increase crop yield.

**Keywords:** biofortification; stress; sustainable agriculture; zinc; zinc-solubilizing bacteria



## 1. Introduction

The majority of soils around the world are lacking in plant nutrients, especially micronutrients, and this deficiency leads to the low productivity of agricultural products. Insufficient nutrition and hunger are the two most serious threats to millions of people in poor countries. The use of important nutrients in soil is regarded as an integral component of agriculture in advanced nations. The nutrition of plants is a crucial aspect of improving general cultivation, including the standard of plant products [1]. Micronutrients such as manganese (Mn), molybdenum (Mo), iron (Fe), boron (B), chloride (Cl), copper (Cu), and zinc (Zn) are essential for livestock, plants, human growth, and development. However, shortages of these nutrients have been recorded in different parts of the world, including Australia, Pakistan, Afghanistan, China, Brazil, Turkey, Africa, Iran, the United States, Iraq, and India, ultimately impacting the soil system and, hence, crop productivity [2].

One of the important elements for optimal plant growth is Zn. Zinc plays an important role in various growth and metabolic aspects, including photosynthesis, sugar synthesis, protein synthesis, fertilization and seed formation, growth regulation, disease resistance, etc. All these processes are hampered by Zn deficiency, which ultimately reduces the productivity potential of crop plants [3]. Both humans and animals need a significant amount of Zn; thus, there is a great likelihood that human health will be impacted in regions where crops are frequently deficient in Zn. This is proven by the fact that fertility issues have become more prevalent in people of these regions over the past few years, and

because the result is much stronger in animals. An absence of pasture, along with poor animal nutrition, affects the regularity, resulting in animal conception irregularities. By examining the quantity of microminerals in feed and fodder samples, numerous studies were carried out to determine the prevalence of micromineral shortages in animals. One element that has been discovered to be severely lacking in several Indian geographic regions is Zn [4]. Although there is more than enough Zn in the soil to support crop growth, plants are not able to uptake it, due to the presence of inaccessible Zn fragments. Numerous factors influence Zn's availability in the soil, including soil texture, pH, soil phosphorus, and meteorological conditions. Exogenous usage of Zn in the form of fertilizers is also unorthodox, due to its quick transformation into inaccessible components and buildup in the soil [5]. Therefore, there is a dire need for some exogenous strategies that can increase the availability and accessibility of Zn to the plants for their optimal growth and development.

Recently, various techniques have been used to reduce Zn deficiency in host plants. Among these, the application of chemical fertilizers is a costly and unsustainable approach, making crops susceptible to illnesses and, over time, reducing the soil's fertility. Moreover, indiscriminate fertilizer use has polluted the soil and water, endangering both human health and plant life. Therefore, the interest of researchers is focused on the use of eco-friendly and economic tools which can improve the availability of nutrients (especially Zn) without harming the environment. The use of Zn-solubilizing bacteria (ZSB) is a low-cost alternative technique for Zn biofortification, providing the optimal sustainable approach to environmentally friendly farming. Because microorganisms must exist for Zn solubilization, enhancing the capacity of microbes to solubilize many different insoluble Zn components can be efficiently used to boost the bioavailability of Zn in crops [2]. ZSB found in the rhizospheric hub, as well as in interior plant tissues, demonstrate their ability to solubilize Zn in many different kinds of ways [6]. ZSB-mediated plant growth is generally a consequence of possibly "direct" or "indirect" plant growth-related strategies. ZSB may affect the plant hormone concentration in plants, as well as improve the acquisition of crucial micronutrients via the accumulation of organic acids and enzymes through direct processes. The uptake of nutrients aided by ZSB may frequently include Zn, Fe, P, N, and K. The indirect mechanism characteristic includes the generation of secondary metabolites, specifically antifungal metabolites, followed by antibacterial chemicals, which might mitigate plant damage caused by phytopathogens (such as soil fungi and bacteria) [7]. Additionally, ZSB have the ability to modulate soil properties, which further enhance the availability of Zn to the plants [8].

Keeping in view the above facts, the current review focuses on the magnitude of Zn deficiency, as well as its availability in the soil and its impact on growth and productivity of plants. Moreover, the study discussed the underlying mechanism of Zn solubilization by ZSB, which could be a sustainable approach to improve the bioeconomy of food crops by increasing their quality and productivity.

## 2. Zinc Bioavailability in Soil

The concentration of Zn in the soil majorly depends on the physicochemical properties of the soil. The major edaphic factors affecting the availability of Zn include pH and redox conditions, content of organic matter, total Zn concentration, and microbial activities in the soil. The activity of Zn is the soil has a direct relationship with increasing proton activity; therefore, the solubility of Zn will always be inversely proportional to the soil pH [9]. For example, the solubility and mobility of Zn were reported to be higher in acidic soil than in alkaline, indicating that the pH of the soil is an important factor for Zn availability [10]. Similarly, soil amended with organic matter displayed higher mobility of metals, including Zn, further revealing the impact of soil properties of Zn availability [11]. The amount of Zn present in the soil can be determined through the geochemical composition, as well as the deterioration, of the primary rock (Table 1). The production and consumption of Zn-rich goods, as well as contamination from the environment, may alter the composition of the

parent rock. The Zn content of the Earth's crust is 78 mg/kg, varying among the parent rocks. Zn concentration, for instance, ranges from 40 to 120 mg/kg in magmatic rocks, 10 to 25 mg/kg in dolomites, 15 to 30 mg/kg in sandstones, and 80 to 120 mg/kg in sedimentary rocks and limestones [12].

**Table 1.** List of Zn mineral ores in soil.

| Mineral Zn | Complexed Zn | Adsorbed Zn |
|---|---|---|
| Smithsonite ($ZnCO_3$) | Manure | $Zn$-$CaCO_3$ |
| Sphalerite ($ZnS$) | Organic | $Zn$-$MgCO_3$ |
| Zincite ($ZnO$) | Residues | $Zn$-$FeO$ |
| Franklinit ($ZnFe_2O_4$) | | $Zn$-$MnO$ |
| Wellemite ($Zn_2SiO_4$) Hopeite [$Zn_3(PO_4)_2 \cdot 4H_2O$] | | |

## 3. Effectiveness of Zn Fertilizers in Soil

Understanding the cause of Zn scarcity can help in planning appropriate actions to increase soil fertility and crop productivity. The use of various fertilizers, the selection of which is determined by their cost, simplicity, economic compatibility, administration method, and environmental acceptability, is now the most prevalent option used to address Zn shortages. Zn fertilizers are often divided into three categories, namely inorganic, natural organic, and synthetic chelate complexes [4]. $ZnO$, $ZnCO_3$, $ZnSO_4$, $Zn_3(PO_4)_2$, and $ZnCl_2$ are examples of inorganic Zn sources. The most efficient, readily available, and least expensive fertilizer applied through soil or foliar is Zn sulfate heptahydrate ($ZnSO_4 \cdot 7H_2O$) [2]. Organic Zn fertilizers are composed of a range of ingredients, the most common of which are Zn phenolate, Zn-EDTA, and Zn lignosulfonate. Fertilizers such as zincated urea, zincated super, and boronated super have demonstrated long-term effectiveness in increasing soil fertility and lowering plant micronutrient deficiency, when paired with micronutrients. Farmers use organic fertilizers less frequently, due to their inefficiency and budgetary issues [13]. To overcome these challenges, scientists are working toward advancing the development of biofertilizers that could increase the soil's soluble Zn concentration without harming the environment [4].

## 4. Physiological Functions of Zn in Plants

Zinc needs to be present in minute but important amounts for numerous essential plant physiological pathways to function properly. Zinc is a structural constituent or regulatory cofactor of many different enzymes and proteins involved in various metabolic pathways in plants; for instance, in photosynthesis and carbohydrate metabolism, protein as well as auxin metabolism, the maintenance of membrane integrity, pollen formation, and resistance to pathogen attack [12,14,15]. Because Zn is required for the activity of a wide range of enzymes, Zn shortages therefore impair protein, carbohydrate, and auxin metabolism, as well as reproductive activities [12,14]. Zinc plays a vital role in maintaining cellular membrane integrity, by modulating the detoxification of reactive oxygen species [16]. Moreover, Zn-deficient plants are more susceptible to root infections like *Fusarium graminearum*, due to greater leakage of carbon-containing chemicals into the rhizosphere [12]. Zinc also safeguards plants against oxidative stress by increasing the activity of antioxidant enzymes such as SOD (superoxide dismutase), POD (peroxidase), CAT (catalase), APX (ascorbate peroxidase), and GR (glutathione reductase) [17]. Furthermore, Zn aids in pollination by influencing the pollen tube development [12]. Zn is also required for the maintenance of living membranes and is also linked to membrane phospholipids as well as sulfhydryl component groups [15]. It can also form tetragonal compounds with cysteine polypeptide chain residues, protecting proteins and lipids from oxidative damage [18]. Zn deficiency in plants is associated with the disruption of normal enzyme action, which, for instance, inhibits photosynthesis, as Zn is a cofactor of carbonic anhydrase, boosting the fixation of $CO_2$ in the chloroplast and consequently the Rubisco enzyme's carboxylation capabilities [18]. Furthermore, a deficiency of Zn ions produces a variety of irregularities in the development of

plants, e.g., dwarfism, chlorosis, and, more specifically, spikelet sterility [15]. Additionally, Zn deficiency has an adverse effect on the quality of harvested crop products, infection induced by pathogen attacks, and plant susceptibility to various abiotic stresses [12].

## 5. Zinc-Solubilizing Bacteria (ZSB) as a Biofertilizer

Biofertilizer is defined as a substance containing microorganisms that are living and, when applied to seed, plant surfaces, or soil, colonize the rhizosphere or the plant and facilitate development through boosting the intake, along with the accessibility, of essential nutrients for the host plant [19,20]. Microbial inoculants have various advantages over chemical alternatives. They are environmentally friendly and sustainable sources of renewable nutrients necessary for soil health and life [21,22]. They also have negative effects on various agricultural diseases that help the plants to resist unfavorable circumstances [19]. In accordance with the ability, they have to acquire nutrients from the soil, fix atmospheric nitrogen, drive nutrient solubilization, and function as biocontrol agents [23]; therefore, various microbiological species are being widely exploited to serve as effective natural fertilizers.

ZSB can help to overcome Zn shortages by turning insoluble Zn into soluble Zn, improving its availability and the efficiency of its uptake by host plants. The selection and inoculation of ZSB, either in pure form or in combination with inexpensive insoluble Zn substances, would reduce the expense of manufacturing the agricultural product [1]. A number of microbes have also been shown to serve an important role with regard to the solubilization of potassium (K), phosphorus (P), iron (Fe), silicates, and Zn in plant roots. Khan et al. [24] reported that Zn-mobilizing plant growth promoting rhizobacteria (PGPR) significantly improved total biomass, harvest index, yield, and Zn content in rice grains, and reduced the symptoms of Zn deficiency. Rehman et al. [25] reported a higher efficacy of ZSB, i.e., *Pseudomonas* sp., in improving the productivity of wheat plants. Abaid-Ullah et al. [26] qualitatively and quantitatively selected 9 out of 50 ZSB on a variety of insoluble Zn ores, including $Zn(CO_3)_2$, $ZnO$, $Zn(PO_4)_3$, and $ZnS$, and recorded the higher bioavailability of Zn in ZSB-inoculated ores. Among the ZSB, *Serratia liquefaciens*, *S. marcescens*, and *B. thuringiensis* FA-2, FA-3, and FA-4 strains outperformed and improved the Zn uptake in grains by 68%, 57% and 46%, respectively. When PGPR strains are inoculated into various plant species, they have been demonstrated to boost the availability and uptake of Zn [19]. In a nutshell, ZSB can increase the bioavailability of Zn to crops by making it soluble, from both organic and inorganic pools of total soil Zn.

## 6. Roles of ZSB in the Biofortification of Crop Plants

### 6.1. Mechanism of Action

Insoluble zincate formation as a result of Zn fertilizer applications is a severe threat to the plant–soil system. Zinc-solubilizing bacteria can be employed as an alternative to Zn supplements, as they can convert insoluble forms of Zn into soluble forms. Moreover, they can improve plant growth and development by breaking down complex Zn molecules into simpler forms, thereby boosting the quantity of Zn accessible to the plants. The capacity of PGPR to dissolve metal salts is critical, because it allows plants to use the mobilized forms. PGPR uses a range of mechanisms to solubilize nutrients in soil; for example, acidification through the production of organic acids, exchange reactions, the manufacturing of metal chelating molecules known as "siderophores" or "chelated ligands", and the involvement of an oxidation–reduction system [4,27] and gluconate or gluconic acid derivatives that consist of 2-keto-gluconic acid and 5-keto-gluconic acid [2], as well as many other organic acids produced by PGPR, are likely the mechanisms by which Fe and Zn are mobilized [28]. Acidification is the most prevalent method that ZSB choose to improve the solubilization and bioavailability of Zn. Sindhu et al. [6] observed that biofertilizer varieties featuring *Pseudomonas* sp., *Agrobacterium* sp., and *Azospirillum lipoferum* released insoluble Zn as a result of the production of the chelating agent ethylenediaminetetraacetic acid, thereby making the Zn accessible to rice. Moreover, ZSB also produce organic acids in the soil,

which act as a reservoir for Zn cations, leading to the lowering of the soil pH around them [29]. The synthesis of gluconate, or derivatives of gluconic acid, such as 2-keto-gluconic acid, 5-keto-gluconic acid, and many other organic acids, by ZSB are likely key components in the procedure of Fe and Zn absorption [4]. Yadav et al. [30] reported that the transformation of insoluble forms of Zn compounds to soluble forms is achieved by *Bacillus* species through the secretion of organic acids, proton extrusion, and the synthesis of chelating ligands; also, ZSB produces gluconic acid and 2-ketogluconic acid, which are the primary acids that regulate Zn solubilization.

### 6.2. Chelation of Zn by Siderophore

Zn-chelating substances raise Zn's bioavailability in the root rhizosphere and are released by the roots of plants and ZSB. The ZSB release a variety of compounds that bind to $Zn^{2+}$ to lessen their interaction within the soil [5]. Siderophores are potent soluble Zn-binding chelating agents. These substances are tiny, high-affinity molecules that are released by bacteria, fungi, and plants. Because of low Zn solubility at high pH levels, these substances are formed by a variety of bacteria, including those that respond to Zn deficiency, which often occurs in neutral to alkaline pH soils. Kumar et al. [31] have identified siderophore-producing microorganisms from the rhizosphere that are members of the following genera: *Bradyrhizobium*, *B. megaterium*, *P. aeruginosa*, *Pseudomonas*, *Serratia*, and *Streptomyces*. Verma et al. [32] identified *Bacillus altitudinis* C7 and *Pseudonocardia alni* M29 as major siderophore producers, which exhibited the potential ability to solubilize Zn. Similarly, Bhatt and Maheshwari [33] reported that *Bacillus megaterium*, a ZSB, was able to enhance plant growth through siderophore production. Similarly, *Serratia* sp. and *Acinetobacter* sp. were also observed to be siderophore-producing ZSB by Othman et al. [34].

### 6.3. Molecular Mechanism of Zn Uptake and Translocation in Plants

Genomes of plants are made from a vast array of genes which exhibit accurate sequences of expression, in accordance with the absorption and transportation of Zn. This mechanism guarantees that all tissues, particularly the edible portions, acquire an adequate quantity of essential nutrients needed to maintain the essential functions of the cell. Certain genes, particularly those belonging to the *ZIP* family, were recently identified in plants and are essential for the transportation and build-up of Zn [35]. Elevated or decreased Zn concentrations affect the way these particular genes show themselves. Various plants were discovered to have upregulated expressions of *ZIP* family genes during a Zn shortage [6]. Ajeesh Krishna et al. [36] reported 16 *ZIP* transporters in different plant parts of rice, comprehending the mechanics of Zn transport. A significant influx transporter was found in the plasma membrane of rice, *ZIP-OsZIP9*, demonstrating its role in Zn uptake [37]. Deshpande et al. [38] reported that the *TaZIP* family of genes are essential for the uptake and movement of Zn in different parts of wheat. Similarly, the expression of *TaZIP* transporters, including *TaZIP3*, *TaZIP5*, *TaZIP6*, *TaZIP7*, and *TaZIP13*, were elevated in the shoot and root of wheat during Zn deficiency [39].

The details of ZSB's role in improving the growth and productivity of host plants are listed in Table 2.

**Table 2.** List depicting the mode of action of various ZSB in various host plants.

| S. No. | Name of ZSB | Host Plants | Mode of Action | References |
|:---:|:---:|:---:|:---:|:---:|
| 1. | *Pantoea dispersa*, *P. agglomerans*, *Pseudomonas fragi*, *Rhizobium* sp., and *E. cloacae* | *Triticum aestivum* | Increased shoot dry weight and Zn uptake and accelerated the bioavailability of Zn. | [40] |
| 2. | *Bacillus* sp. | *Zea mays* | Promoted root and shoot length, dry and fresh weight, transpiration rate, and chlorophyll content. | [41,42] |

**Table 2.** *Cont.*

| S. No. | Name of ZSB | Host Plants | Mode of Action | References |
|--------|-------------|-------------|----------------|------------|
| 3. | *Bacillus* sp. | *Oryza sativa* | Higher photosynthetic rate, transpiration rate, stomatal conductance, and carbonic hydrase activity, as well as reduced electrolytic leakage. | [43] |
| 4. | *Trichoderma harzianum* and *Bacillus amyloliquefaciens* | *Triticum aestivum* | Upregulated the expression of ZIP transporters, caused more plant growth, and improved Zn fortification. | [44] |
| 5. | *Bacillus aryabahttai* | *Oryza sativa* | Improved plant biometrics, especially grain yield. | [45] |
| 6. | *Ralstonia picketti, Pseudomonas aeruginosa, Klebsiella pneumoniae* and *Burkholderia cepacia* | *Oryza sativa* | Increased Zn biofortification, growth, and Zn bioaccessibility to the plants. | [46] |
| 7. | *Burkholderia* and *Acinetobacter* | *Oryza sativa* | Improved dry matter production, the number of panicles, grain and straw yield, and Zn uptake. | [47] |
| 8. | *Ochrobactrum intermedium, Paenibacillus polymyxa, Bacillus cereus, Stenotrophomonas maltophili, Streptomyces,* and *Arthrobacter globiformi* | *Cicer arietinum* | Increased availability of Zn, increased nitrogen (N) and P content in grain, and increased Zn content in shoot, roots, and grains. | [48] |
| 9. | *Burkholderia cepacia* and *Acinetobacter baumannii* | *Zea mays* | Improved plant height, root length, and Zn uptake. | [49] |
| 10. | *Pseudomonas* and *Bacillus* spp. | *Zea mays* | Higher plant growth and increased N, K, Mn, and Zn uptake. | [50,51] |
| 11. | *Pseudomonas protegens* | *Cicer arietinum* | Enhanced shoot and root growth as well as Zn uptake. | [52] |
| 12. | *Pantoea* sp., *Klebsiella* sp., *Brevibacterium* sp., *Klebsiella* sp., *Acinetobacter* sp., *Alcaligenes* sp. NCCP-650, *Citrobacter* sp., *Exiguobacterium* sp., *Raoultella* sp., and *Acinetobacter* sp. | *Triticum aestivum* | Improved dry weights, fresh weights, and Zn acquisition. | [53] |
| 13. | *Exiguobacterium aurantiacum* | *Triticum aestivum* | Increased nutritional quality of seeds by enhancing the accumulation of Zn, Fe, N, P, and K. | [54] |
| 14. | *Enterobacter cloacae* | *Oryza sativa* | Upregulated the expression of *ZIP* genes and increased the accumulation of Zn in root and shoot. | [55] |
| 15. | *Neisseria, Staphylococcus cocci, Escherichia coli,* and *Bacillus* sp. | *Vigna radiata* | Improved plant growth attributes including root and shoot length and fresh and dry weight. | [56] |

**Table 2.** *Cont.*

| S. No. | Name of ZSB | Host Plants | Mode of Action | References |
|---|---|---|---|---|
| **16.** | *Bacillus altitudinis* | *Cicer arietinum* | Improved growth attributes and higher Zn uptake. | [57] |
| **17.** | *Enterobacter* sp. | *Cicer arietinum* | Improved yield, bioavailability of Zn, and grain quality. | [58] |
| **18.** | *Bacillus aryabhattai* | *Triticum aestivum*, *Glycine max* | Reduced soil pH, increased the production of total organic acid, and improved soil enzymatic activities. | [59] |
| **19.** | *Acinetobacter calcoaceticus*, *Bacillus proteolyticus* and *Stenotrophomonas pavanii* | *Zea mays* | Higher Zn content and plant dry weight. | [60] |
| **20.** | *Serratia* sp. | *Zea mays* | Increased peroxidase, superoxide dismutase, catalase, and polyphenol activity. | [61] |
| **21.** | *Streptomyces* spp. | *Glycine max* | Increased root and shoot length, dry weight of plants, and number of pods. | [62] |
| **22.** | *Bacillus* spp. | *Triticum aestivum* | Enhanced nutrient use efficacy, growth, yield, and Zn biofortification. | [63] |

*6.4. Zn-Assisted Biofortification*

The role of ZSB in the amendment of synthetic Zn fertilizers in the soil and their transformation into an inaccessible substance, known as a Zn compound, exacerbates the problem of Zn immobility from soil to plant system, and this issue is capable of being solved by using ZSB inoculants [6]. The use of ZSB as a bioinoculant is an affordable way of biofortifying food crops with Zn. Moreover, the utilization of ZSB, which has several plant growth-promoting qualities, represents an innovative approach towards generating sustainable bio-fortified crops [6]. ZSB, living in the rhizosphere, colonized the host plants effectively, allowing their functioning as a supplementary partner of the plant root, for the improved absorption of nutrients, by solubilizing the complex or unavailable form of Zn in soils [64]. Rhizobacteria are widely recognized microorganisms that live and colonize in the rhizosphere and exhibit a variety of plant-growth-related characteristics, including phosphate and potassium solubilization, exopolysaccharide and siderophore production, phytohormones synthesis (gibberellins, auxin cytokinins, etc.), and HCN production [65]. According to previous research, various strains of ZSB have been reported to play vital roles in Zn biofortification of some food crops. For instance, *Burkholderia cepacia* improved growth, grain yield, dry weight, and Zn acquisition in rice [66], *Bacillus altitudinis* in chickpea [48,67], *B. tequilensis* in wheat [62,68], and *Pseudomonas* spp. in tomato [69]. Similarly, *B. aryabhattai*, as well as *B. subtilis*, enhanced cob dry weight, cob length, and grain yield in maize plants [70], *R. tropici* and *B. subtilis* improved dry matter and grain yield in common bean [71], and *Pseudomonas plecoglossicida*, as well as *Brevibacterium antiqunum*, increased plant height, dry biomass, and productivity of pigeon pea [67]. Moreover, Zn-solubilizing endophytes acted as biofortifying agents, to improve Zn localization in the eatable part of rice as well as chickpeas [57] and wheat [25]. Some ZSB, such as *Bacillus* sp. [72], *Pantoea dispersa*, *Pseudomonas fragi*, *Enterobacter cloacae*, *Pantoea agglomerans*, *Rhizobium* sp., *Acinetobacter*, and *Burkholderia* have all been employed effectively as bio-inoculants for bacteria-assisted biofortification [6,73].

### 6.5. ZSB as a Stress Alleviator

Extrinsic factors that have an adverse effect on growth and development of plants are commonly referred as "stresses" [74,75]. Signals from stress, including drought, heat, salinity, herbivory, and pathogens, are known to be perceived and responded by all plants [76–78]. Plant existence against diverse abiotic and biotic challenges is based on their timely preparedness in modifying their inherent tolerance mechanisms to mitigate the effects of environmental stresses [79–81]. However, under severe stressed conditions, plants are not able to overcome these unfavourable circumstances; therefore, exogenous stress-alleviative and eco-friendly strategies are required (Figure 1). Because of the varied roles of PGPR, including ZSB, they are widely used to decrease abiotic stresses produced by climate change [82]. ZSB has the ability to make plants resistant to certain abiotic stressors; therefore, improving the density of ZSB in the plant rhizosphere could be an effective alternative approach to increase the growth and productivity of host plants [24]. Barnwal et al. [83] reported that *Arthrobacter protophormiae*, as well as *Dietzia natronolimnaea*, improved salt tolerance, whereas *Bacillus subtilis* increased the drought resilience in wheat plants by modifying the levels of phytohormone. Jha and Subramanian, in [84], when investigating rice plants treated with *Pseudomonas pseudoalcaligenes* and *Bacillus pumilus*, recorded a higher expression of stress-related genes and an increased level of osmoprotectants under salinity-stressed conditions. Potato plants inoculated with *Methylobacterium* sp. displayed an increased number of lateral roots and leaves, a higher rosette diameter and improved tolerance against salinity and different fungal pathogens [85]. The activity of different antioxidant enzymes (superoxide dismutase, catalase, and peroxidase) was upregulated in ZSB-inoculated wheat plants and, thereby, displayed better resistance to salt stress. *Pseudomonas fluorescens*- and *P. poae*-boosted growth of petunia plants under drought and low-nutrient environments [86]. *Pseudomonas fluorescens* and *Bacillus subtilis* negated the effects of salinity by enhancing the accumulation of proline (an active osmolyte) [87]. On the same line, Orozco-Mosqueda et al. [88] revealed that *Pseudomonas* sp. protected tomato plants against salt stress. El-Esawi et al. [89] reported that *Azospirillum lipoferum* reduced the negative effects of salt stress by modulating osmolytes synthesis, antioxidant enzymes, and the expression of stress-related genes in chickpea.

**Abiotic/biotic stresses:**
- Impaired plant growth
- Decreased photosynthesis
- Reduced nutrient acquisition
- Higher accumulation of ROS
- Disruption of phytohormone level
- Increased membrane permeability
- Enhanced disease susceptibility
- Altered soil physico-chemical properties

● Zn-Solubilising Bacteria

**"ZSB–Plants" interaction:**
- Improved growth and productivity
- Higher uptake of nutrients
- Increased phytohormones level
- Enhanced osmolytes accumulation
- Strengthened the antioxidant defense system
- Modulated the expression of stress-responsive genes
- Induced systemic resistance

**"ZSB–Soil" interaction:**
- Improved synthesis of organic acids
- Increased chelation
- Modulated soil pH
- Enhanced siderophore production
- Higher bioavailability of Zn

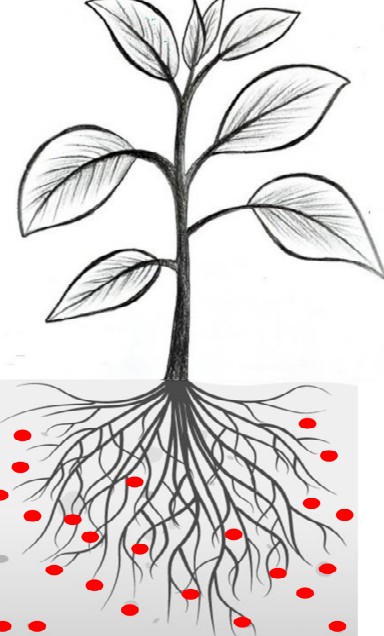

**Figure 1.** Interactive roles of ZSB in modulating growth and rhizospheric condition of plants under abiotic/biotic constraints.

Besides abiotic factors, plants are exposed to a wide range of biotic agents, including microorganisms such as viruses, viroids, bacteria, fungus, nematodes, etc., which can cause biotic stress and ultimately, reduce agricultural productivity (Figure 1). ZSB exhibit diverse direct and indirect mechanisms to suppress the diseases caused by various pathogens, such as the synthesis of secondary metabolites, antioxidants, hormones, cell wall-degrading enzymes, etc. [76,90]. Many ZSB produce antagonistic low-molecular-weight chemical molecules known as antibiotics, which are highly effective in inhibiting phytopathogen growth. For example, *Bacillus* sp. have the ability to produce antibiotics like polymyxin, circulin, and colistin [76]. Ali et al. [91] reported that *B. subtilis* synthesized surfactins, fengycin, and iturin, whereas *Pseudomonas aeruginosa*, *P. stutzeri*, *P. fluorescens*, and *P. putida* produced phenazine, rhamnolipids, pyochelin, violacein, and pyoverdines, which were efficient against different fungal phytopathogens. Research conducted by Fernandez et al. [92] showed that *Pseudomonas fluorescens* produced two different kinds of S-type bacteriocins, i.e., colicin and tailocins, which are phage-tail-like bacteriocins. The synthesis of lytic enzymes, like protease, chitinase, cellulase, b-1,3-glucanase, etc., is linked to the fact that some ZSB have inhibitory effects, in opposition to phytopathogens [93]. Several studies have proven that rhizospheric bacteria produce lytic enzymes that break down the cell wall of plant root pathogens, for instance *Rhizoctonia solani* and *Fusarium oxysporum*, thereby leading to cell death [91]. Cheng et al. [94] reported the inhibitory properties of *Bacillus megaterium*, *B. cereus*, and *Bacillus* sp. against maize rot pathogen. Non-pathogenic PGPR organisms found in soil not only stimulate plant development but can also cause systemic resistance, known as induced systemic resistance (ISR). Many plants develop systemic resistance (both SAR—systemic acquired resistance—and ISR) to various biotic stresses as a result of PGPR, which defends against pathogen attacks [95]. *Pseudomonas putida* and *Bacillus subtilis* PGPR strains imparted systemic resistance in *Vigna radiata* grown under disease-prone environments [96]. *Bacillus amyloliquefaciens* improved tomato plants' susceptibility to yellow leaf curl virus disease by upregulating the expression of pathogenesis-related (PR) genes and improving the activity of b-1,3 glucanase, phenylalanine ammonia lyase, peroxidase, and polyphenol oxidase, as well as chitinase in the leaves [97]. In a nutshell, soil rhizobacteria have the potential to mitigate the impact of various environmental constraints (abiotic and biotic) in a sustainable manner (Table 3).

**Table 3.** Differential ability of ZSB in mitigating the impact of abiotic and biotic stresses in various crop plants.

| Stress | ZSB | Plant | Mechanism of Action | References |
|---|---|---|---|---|
| Salinity | *Bacillus amyloliquefaciens* B-16 | *Triticum aestivum* L. | Increased uptake and translocation of potassium and calcium. | [98] |
| | *Bacillus pumilus* and *Pseudomonas pseudoalcaligenes* | *Oryza sativa* | Improved chlorophyll, carotenoids, and antioxidant enzymes activity. | [99] |
| | *Pantoea agglomerans* R1 and *Pseudomonas fragi* R4 | *Phaseolus vulgaris* | Higher chlorophyll, carotenoid, and osmoprotectants levels, and improved antioxidative enzymes activity. | [100] |
| | *Bacillus* spp. | *Triticum aestivum* L. | Increased plant growth parameters and Zn content in shoots as well as grains. | [101] |
| Drought | *Bacillus* spp. | *Zea mays* | Improved physiological and biochemical traits, alongside reduced antioxidant enzyme activity. | [102] |
| | *Azotobacter* | *Zea mays* | Enhanced plant growth. | [103] |

**Table 3.** *Cont.*

| Stress | ZSB | Plant | Mechanism of Action | References |
|---|---|---|---|---|
| Heavy metals | *Serratia* spp. | *Zea mays* | Improved plant growth parameter and antioxidant enzyme activity. | [104] |
| | *Lysinibacillus* spp. | *Zea mays* L. | Increased chlorophyll a and b, proline, total phenol, and ascorbic acid content. | [105] |
| | *Burkholderia vietnamiensis* and *Burkholderia seminalis* | *Oryza sativa* | Induced the production of indole acetic acid (IAA) and the solubilization of potassium and phosphate. | [106] |
| | *Serratia* sp. | *Zea mays* | Enhanced shoot length, root length, and total chlorophyll content. | [107] |
| Temperature | *Stenotrophomonas* | *Zea mays* | Increased carbohydrates, auxins, and chlorophyll contents, and imparted heat stress resilience. | [108] |
| | *L. fusiformis* and *L. sphaericus* | *Zea mays* | Improved lignin content, cell viability, osmolytes (proline, glycine betaine, and soluble sugars) accumulation, total phenols and 1-aminocyclopropane-1-carboxylic acid (ACC) contents, and upregulated the antioxidant defense system. | [109] |
| Disease | *Bacillus* sp. and *Bacillus cereus* | *Oryza sativa* | Suppressed the growth of *Pyricularia oryzae* and *Fusarium moniliforme*, and increased the yield. | [110] |
| | *T. lixii* | *Solanum lycopersicum* | Reduced *Fusarium* wilt and early blight severity. | [111] |
| | *B. pumilus* | *Oryza sativa* | Inhibited fungal growth and reduced brown spot disease. | [112] |

## 7. Conclusions and Future Aspects

Current agricultural practices heavily depend on chemical fertilizers to boost crop output, prioritizing macronutrients and neglecting micronutrients like Zn. This imbalance leads to Zn deficiencies in plants, ultimately affecting their growth and productivity. Efforts to address this issue through fortification and supplementation are expensive and labor-intensive, thereby limiting their success. A promising alternative to overcome this problem is the use of ZSB, in order to reduce the use of commercial fertilizers. The present review discusses the role of ZSB in improving Zn bioavailability in soil and its uptake by plants. ZSB supplementation enhances growth, as well as yield, and biofortifies the crops with Zn in an effective, economical, and eco-friendly manner. ZSB not only addresses Zn deficiencies but also improves the uptake of other essential nutrients like phosphorus, nitrogen, potassium, and iron under adverse environmental circumstances. Moreover, ZSB regulates plant pathogenic microorganisms, contributing to overall soil health and fertility. Concisely, the inoculation of ZSB can be used as an environment-friendly approach for improving plant development and soil health in sustainable agriculture. However, more research work is required for the isolation and identification of suitable ZSB species that can

provide maximum benefit to the host by improving the grain Zn content, thereby fighting against hidden hunger.

**Author Contributions:** S.S. wrote the first draft of this manuscript. R.C. helped with the literature review, while the authors A.S. and A.B. critically revised the manuscript. All authors have read and agreed to the published version of the manuscript.

**Funding:** This research received no external funding.

**Data Availability Statement:** Data are contained within the article.

**Conflicts of Interest:** The authors declare no conflicts of interest.

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
