# Peer review of "Harnessing the Power of Zinc-Solubilizing Bacteria: A Catalyst for a Sustainable Agrosystem"

_2674-1334, doi:10.3390/bacteria3010002_

Round 1
Reviewer 1 Report
Comments and Suggestions for Authors
Harnessing the Power of Zinc-Solubilizing Bacteria: A Catalyst 2 for Sustainable Agrosystem
This manuscript proved the synergistic impact of Zinc-Solubilizing Bacteria on plant growth and stress tolerance. But some suggestions are given to improve this paper further, it mentioned below
Ø Once a word is abbreviated, it is desirable to use it (abbreviation) consistently throughout the text in place of the original word. As for example ‘Zn’ for the word ‘Zinc’. Moreover never start a sentence with abbreviated form
Ø Remove additional space between words
Ø Page 2 line 93-93, restructure the sentence
Ø ‘Abaid-Ullah et al. [26] qualitatively and quantitatively selected 9 out of 50 ZSB on a variety of insoluble Zn ores, including Zn(CO3)2, ZnO, Zn(PO4)3 and ZnS, and recorded higher bioavailability of Zn in ZSB inoculated ores’’. Which are the high performance bacteria?
Ø Expand ‘’ZSB Assisted Biofortification’’ with appropriate subheadings, a detailed view on siderophores" is really essential. Include molecular mechanisms too
Ø Include a table under the section ‘’ ZSB as a Stress Alleviator’’
Ø Add two figures indicating the impact of ZSB in plant growth as well a stress tolerance
Ø Make sure the uniformity in the pattern of references in the list
Ø Modify the section Conclusions and Future Aspects by concise the conclusion and including future aspects.
Ø Rewrite line no 234 in page no 8
Ø Add recent references like
Janeeshma, E., & Puthur, J. T. (2020). Direct and indirect influence of arbuscular mycorrhizae on enhancing metal tolerance of plants. Archives of microbiology, 202(1), 1-16. (IF 2.8)
Janeeshma, E., Rajan, V. K., & Puthur, J. T. (2021). Spectral variations associated with anthocyanin accumulation; an apt tool to evaluate zinc stress in Zea mays L. Chemistry and Ecology, 37(1), 32-49. (IF 2.5)
Comments on the Quality of English Language
Have to revise
Author Response
Reply to reviewer 1
This manuscript proved the synergistic impact of Zinc-Solubilizing Bacteria on plant growth and stress tolerance. But some suggestions are given to improve this paper further, it mentioned below
Query: Once a word is abbreviated, it is desirable to use it (abbreviation) consistently throughout the text in place of the original word. As for example ‘Zn’ for the word ‘Zinc’. Moreover, never start a sentence with abbreviated form.
Reply: Thanks for your valuable suggestion. We have revised the manuscript thoroughly and all the desired changes have been incorporated.
Query: Remove additional space between words
Reply: Modified.
Query: Page 2 line 93-93, restructure the sentence.
Reply: Sentence modified.
Query: Abaid-Ullah et al. [26] qualitatively and quantitatively selected 9 out of 50 ZSB on a variety of insoluble Zn ores, including Zn(CO3)2, ZnO, Zn(PO4)3 and ZnS, and recorded higher bioavailability of Zn in ZSB inoculated ores’’. Which are the high performance bacteria?
Reply: The sentence has been added.
Query: Expand ‘’ZSB Assisted Biofortification’’ with appropriate subheadings, a detailed view on siderophores" is really essential. Include molecular mechanisms too/
Reply: Added and highlighted in the text.
Query: Include a table under the section ‘’ ZSB as a Stress Alleviator’’
Reply: Table 2 has been added in the manuscript.
Query: Add two figures indicating the impact of ZSB in plant growth as well a stress tolerance.
Reply: Thanks for your valuable suggestion. Instead of two figures, we have summarized the impact of abiotic/biotic stress on plants as well as soil attributes and beneficial role of ZSB in imparting stress resilience to the host plants in a single figure. The figure is also incorporated in the main manuscript.
Query: Make sure the uniformity in the pattern of references in the list.
Reply: Reference list has been checked and modified.
Query: Modify the section: Conclusions and Future Aspects by concise the conclusion and including future aspects.
Reply: Modified.
Query: Rewrite line no 234 in page no 8
Reply: Modification done.
Query: Add recent references like
Janeeshma, E., & Puthur, J. T. (2020). Direct and indirect influence of arbuscular mycorrhizae on enhancing metal tolerance of plants. Archives of microbiology, 202(1), 1-16. (IF 2.8)
Janeeshma, E., Rajan, V. K., & Puthur, J. T. (2021). Spectral variations associated with anthocyanin accumulation; an apt tool to evaluate zinc stress in Zea mays L. Chemistry and Ecology, 37(1), 32-49. (IF 2.5)
Reply: The references have been added in the manuscript.

Reviewer 2 Report
Comments and Suggestions for Authors
Dear authors,
Your paper is very interesting. I am asking you to technically harmonize the paper with the journal's requirements.
Best Regards.
Author Response
Reply to reviewer 2
Comments and Suggestions for Authors
Dear authors, your paper is very interesting. I am asking you to technically harmonize the paper with the journal's requirements.
Reply: Thank you very much for your response. We have revised the manuscript as per the requirements of the journal.
Reviewer 3 Report
Comments and Suggestions for Authors
comments

Comments on the Quality of English LanguageAuthor Response
Reply to reviewer 3
- Query: Could you add point on zinc Solubilization Mechanism in details, if it’s possible please??
- Reply: The mechanism of zinc solubilisation has been added at the suitable place and highlighted in the manuscript.
- Query: In recent studies utilized gene expression analysis to illustrate that microbial biostimulants with zinc-solubilizing capabilities, in conjunction with other microbes that promote plant growth, impact the expression patterns of genes belonging to the zinc-regulated transporter family. Consequently, these biostimulants play a crucial role in facilitating the transport of zinc to different parts of the plant. So the authors must describe this role and gave examples on this point if it’s possible please.
- Reply: The necessary changes have been made.
- Query: What about (Quantitative assessments of zinc solubilization) describe in details
- Reply: Suitable studies pertaining to quantitative estimations of Zn have been added and discussed at appropriate places in the manuscript.
- Query: It would be better if you add graphical abstract
- Reply: Thanks for your valuable suggestion. Instead of graphical abstract, the figure related the role of ZSB-plant/soil interaction has been incorporated in the manuscript.
- Query: Explain the behavior of different types of bacteria and their effect on dissolving zinc in the presence of some other metal elements
- Reply: To justify the differential effectiveness of ZSB in the present of other metals as well as other abiotic/biotic stresses, Table 2 has been added in the manuscript.
- Query: The review lacks novelty.
- Reply: To the best of my knowledge, this review article is an attempt to collect all the relevant literature pertaining to detailed aspects of ZSB promoting sustainable agriculture, leading to nutritional food security. Moreover, the review also highlights the importance of ZSB as an efficient strategy to impart stress resilience to crop plants under stressful conditions.
- Query: Grammatical, punctuation; syntax errors for example. There are much grammatical errors in all review. Please check and correct.
- Reply: In the light of above comments, the authors have revised the manuscript thoroughly and all the necessary changes have been incorporated.
Round 2
Reviewer 1 Report
Comments and Suggestions for Authors
Included all the comments, and it is acceptable
Comments on the Quality of English Languagegood
Reviewer 3 Report
Comments and Suggestions for Authors
accept
